# Outcome domains and outcome measures used in studies assessing the effectiveness of interventions to manage non-respiratory sleep disturbances in children with neurodisabilities: a systematic review

Catriona McDaid,[1] Adwoa Parker,[1] Arabella Scantlebury,[1] Caroline Fairhurst,[1] Vicky Dawson,[2] Heather Elphick,[3] Catherine Hewitt,[1] Gemma Spiers,[4] Megan Thomas,[5] Bryony Beresford[6]

For numbered affiliations see end of article.

**Correspondence to**
Dr Catriona McDaid;
catriona.mcdaid@york.ac.uk

## ABSTRACT

**Objectives** To assess whether a core outcome set is required for studies evaluating the effectiveness of interventions for non-respiratory sleep disturbances in children with neurodisabilities.

**Design** Survey of outcome measures used in primary studies identified by a systematic review.

**Data sources** ASSIA, CENTRAL, Cochrane Database of Systematic Reviews, Conference Proceedings Citation Index, CINAHL, DARE, Embase, HMIC, MEDLINE, MEDLINE In-Process, PsycINFO, Science Citation Index, Social Care Online, Social Policy & Practice, ClinicalTrials.gov, WHO International Clinical Trials Registry Platform and the UK Clinical Trials Gateway were searched up to February 2017.

**Eligibility criteria** Studies evaluating pharmacological or non-pharmacological interventions for children (≤18 years old) with a neurodisability and experiencing non-respiratory sleep disturbance.

**Data extraction and synthesis** Outcome measures were listed from each study and categorised into domains.

**Results** Thirty-nine studies assessed five core outcome areas: child sleep, other child outcomes, parent outcomes, adverse events and process measures. There were 54 different measures of child sleep across five domains: global measures; sleep initiation; maintenance; scheduling; and other outcomes. Fifteen non-pharmacological (58%) and four pharmacological studies (31%) reported child outcomes other than sleep using 29 different measures. One pharmacological and 14 non-pharmacological (54%) studies reported parent outcomes (17 different measures). Eleven melatonin studies (85%) recorded adverse events, with variation in how data were collected and reported. One non-pharmacological study reported an explicit method of collecting on adverse events. Several process measures were reported, related to adherence, feasibility of delivery, acceptability and experiences of receiving the intervention.

**Conclusions** There is a lack of consistency between studies in the outcome measures used to assess the effectiveness of interventions for non-respiratory sleep disturbances in children with neurodisabilities. A minimum core outcome set, with international consensus, should be developed in consultation with parents, children and young people, and those involved in supporting families.

**PROSPERO registration number** CRD42016034067

### Strengths and limitations of this study

► Includes multiple interventions and populations thus providing a comprehensive overview of outcome measurement in this field.
► Provides data to inform the first stage of development of a core outcome set.
► The study provides efficient use of data from an existing review but some outcomes that were outside the scope of our effectiveness review may have been missed.

## INTRODUCTION

Child sleep problems are associated with poorer educational outcomes and daytime behaviour difficulties[1] as well as having a negative impact on parents, such as heightened levels of parental stress and irritability.[2] Sleep disturbance can impact on all members of the family and it is often the parents' own poor sleep quality which leads to them seeking help with their child's sleep.[3] Sleep disturbances are more common and severe in children with neurodisabilities compared with typically developing children.[4 5] A recent national research prioritisation exercise ranked management of sleep disturbance for this group of children in the top ten research priorities.[6]

In response to a commissioned call from the UK National Institute for Health Research (NIHR), we undertook a systematic review, to evaluate the effectiveness of pharmacological and non-pharmacological interventions for children with neurodisabilities experiencing non-respiratory sleep disturbances. The NIHR commissioning brief requested a broad systematic review to 'take stock' of the current available evidence in order to inform future research.

One of the challenges we faced while undertaking the review was the diversity of outcomes assessed across the included studies, possibly a reflection of the fact that there is no agreed set of outcomes recommended for use in trials assessing the effectiveness of interventions in this field of research. Based on searches of the COMET database (http://www.comet-initiative.org/ (accessed 21 June 2017)) there is no core outcome set in relation to children with neurodisabilities or typically developing children who experience non-respiratory sleep disturbances. A core outcome set is an agreed minimum, standardised set of outcomes that should be measured and reported in randomised controlled trials (RCTs) for a specific clinical area.[7] They are required to maximise comparability across studies in a specific field in order to facilitate quantitative and narrative synthesis, reduce selective outcome reporting in studies of effectiveness and increase the relevance of results from primary studies and systematic reviews.[7] An important first step, if there is not a core outcome set in a field, is to establish whether one is needed.[8] The breadth of our review, covering both pharmacological and non-pharmacological interventions provided an opportunity to explore whether a core outcome set is required.

The results of the systematic review of effectiveness are available separately (www.journalslibrary.nihr.ac.uk/programmes/hta/1421202/#/).[9–11] The aim of this paper is to summarise the outcome measures used in the RCTs and other effectiveness studies evaluating pharmacological and non-pharmacological interventions for the management of non-respiratory sleep disturbances in children with neurodisabilities and make recommendations about whether a core outcome set is required.

## METHODS
We used the included studies from our systematic review of treatment effectiveness which was based on a prospectively registered review protocol. A summary of the methods relevant to this paper is provided.

### Databases searched
We searched the following without language restrictions up to February 2017: Applied Social Science Abstracts & Indexes (ASSIA), The Cochrane Central Register of Controlled Trials (CENTRAL), Cochrane Database of Systematic Reviews (CDSR), Conference Proceedings Citation Index, Cumulative Index to Nursing & Allied Health (CINAHL), Database of Abstracts of Reviews of Effects (DARE), Embase, Health Management Information Consortium (HMIC), MEDLINE, MEDLINE In-Process, PsycINFO, Science Citation Index, Social Care Online, Social Policy & Practice, ClinicalTrials.gov, WHO International Clinical Trials Registry Platform and the UK Clinical Trials Gateway. The search strategy for ASSIA is in online supplementary file 1. The search results were loaded into EndNote bibliographic software (V.17.0.2.7390, Clarivate Analytics (formerly Thomas Reuters), Philadelphia, PA, USA).

### Study selection
One researcher screened titles only to remove obviously irrelevant records (10% checked by a second researcher), two researchers then independently screened abstracts and ordered potentially relevant papers, the full papers were screened independently by two researchers. We included studies of children and young people (≤18 years old) with neurodisability, experiencing non-respiratory sleep disturbances, which evaluated pharmacological (melatonin, clonidine and antihistamines) or non-pharmacological Interventions (eg, behavioural, cognitive-behavioural, self-help resources, complementary therapies). We used the Morris *et al* definition of neurodisability as 'congenital or acquired long-term conditions that are attributed to impairment of the brain and/or neuromuscular system and create functional limitations'.[12] A best available evidence approach was used with non-randomised controlled studies and before and after studies included where RCT evidence was not available.

### Outcomes
A broad range of outcomes were of interest:

*Primary outcomes:* child sleep-related outcomes (including parent/carer and child reported outcomes (eg, using sleep diaries and actigraphy), parent sleep-related outcomes (eg, quality of sleep) and measures of perceived parenting confidence, efficacy or understanding of sleep/sleep management (which can be a specific focus of parent training/behavioural interventions which seek to change how parents manage sleep disturbance).

*Secondary outcomes:* child-related quality of life, daytime behaviour and cognition; parent/carer outcomes such as quality of life and well-being, physical well-being, mental well-being; family functioning; and adverse events.

### Data extraction and synthesis
For each study the specific outcome measures used were extracted with details of how the outcome was defined or described by the author (where reported) and, where relevant, the method of assessment, for example, child/parent reported outcome measure, sleep diaries, actigraphy, polysomnography. Data were extracted by one researcher and checked by a second.

Outcomes were tabulated and grouped into core areas, then by outcome domain and specific outcome measure used. This was undertaken by a single researcher and

discussed with the wider team. The data were described in a narrative synthesis grouped by the categories.

## Patient and public involvement

There was patient and public involvement in the decision by NIHR to commission the overall project focusing on sleep disturbances in children with neurodisabilities. Three parents of children with neurodisabilities, identified from an existing parent consultation group agreed to be involved with the project. They were invited to project team meetings, which took place three times during the project, and were consulted via email regarding the implications of the review findings. Their contributions provided useful contextual information. In addition, a member of a sleep charity, which supports parents, was involved at all stages of the project and is a co-author (VD) of the publication.

## RESULTS

### Overview of included studies

After removal of duplicates 15 745 records were screened and 387 full-text records assessed for eligibility. Thirty-nine studies (reported across 64 articles) were included: 13 studies of melatonin, published between 1996 and 2012 and 26 studies of parent-directed, environmental, dietary and complementary medicine interventions, published between 1991 and 2016 (table 1). The latter group of studies are referred to as non-pharmacological for convenience. The studies were most commonly undertaken in the UK (36%) and USA (26%) followed by Australia (15%) and Canada (13%). Single studies were reported from the Netherlands, Italy, Israel and China. Study designs encompassed RCTs (both parallel-group and crossover), and for the non-pharmacological interventions, controlled before and after studies and one-arm before and after studies were also included.

The studies assessed five core outcome areas: child sleep, other child outcomes, parent-related, adverse events and process outcomes. All of the studies (n=39) reported at least one child sleep-related outcome: 19 (49%) reported other types of child outcomes such as daytime behaviour and cognition; and 14 reported parent/carer outcomes (36%) (table 1).

For the non-pharmacological studies, the first outcome measurement time points ranged from immediately post intervention to 2 months post intervention and five measured additional follow-up time points. All the melatonin studies had a single follow-up time point immediately following the completion of the intervention, which ranged from 10 days to 12 weeks postrandomisation.

### Child sleep outcomes

Table 2 reports the child sleep outcome measures used by at least one study. Fifty-four different measures of child related sleep outcomes were reported. There were five domains: global assessments of sleep (n=17 outcome measures); sleep initiation (n=10), sleep maintenance

(n=18), sleep scheduling (n=4) and other outcomes (n=5).

In the melatonin studies, total sleep time (TST) was the most commonly reported global sleep outcome, reported by 12 of the 13 included studies (92%). Four reported TST derived from parent-completed sleep diaries and actigraphy[13–16]; three reported actigraphy derived data only[17–19]; six sleep diary derived data.[20–25] The most common way to measure the sleep initiation domain was sleep onset latency (SOL, 85%); and within the sleep maintenance domain was night awakening (46%). As with TST, these outcomes were assessed using both actigraphy and sleep diary (reporting results of both or actigraphy only) or using sleep diary only. Sleep scheduling and parent-reported global measures of sleep were rarely reported in the melatonin studies (table 2).

For the non-pharmacological studies, the sleep outcome measures used were more disparate with only one measure reported by more than half the studies. The most commonly reported global measure of sleep (and the most commonly reported overall) was the parent-reported Child Sleep Habits Questionnaire (CSHQ; 58%). TST calculated from actigraphy or parent-reported sleep diary data was much less commonly used (42%) compared with melatonin studies. Six studies reported actigraphy derived data,[26–30] three reported parent completed sleep diary derived data,[31–33] one reported actigraphy and diary derived data,[34] and one reported data based on 'semiweekly' phone calls to parents.[35] For the sleep initiation domain, the most commonly reported was SOL (39%); for sleep maintenance it was number of night awakenings (15%) and Wake After Sleep Onset (WASO; 19%); sleep scheduling was rarely reported. There was some overlap between pharmacological and non-pharmacological studies in the measures used, most commonly TST, sleep efficiency and SOL (table 2).

Even where studies were measuring the same outcome, there was variation in how this was measured, though often the method used was not specified. For example, in terms of wake time, one study used actigraphy and defined the outcome as the last epoch of actigraphically assessed immobility before the start of a 10 min consecutive period of activity around the time of leaving bed[18]; two used parent-reported wake time using a sleep diary (no further definition provided)[16 36]; and one study used a momentary time sampling observation of children who were in-patients.[37] WASO and night awakening both assess sleep maintenance, but different approaches to measurement were used. One study used actigraphy to measure night awakenings (definition not provided)[38]; one used a sleep diary (defining night awakenings as the number of awakenings per week that parents were aware of)[33]; and the other two did not provide a definition.[35 36] All studies reporting WASO used actigraphy, two of which defined WASO as a sum of the all wake epochs during the sleep period (total time awake during the night excluding the SOL period)[28 39] (two did not specify how it was defined[27 29]). Where SOL was defined, there was also

**Table 1** Summary of included studies

| Study | Year Country | Type of neurodisability | Intervention | Study design | Child outcomes Sleep | Other | Parent outcomes | Adverse events |
|---|---|---|---|---|---|---|---|---|
| **Melatonin** | | | | | | | | |
| Appleton et al[13] | 2012 UK | DD alone or with other condition | Melatonin | Parallel RCT | X | X | X | X |
| Camfield et al[20] | 1996 Canada | Mixed | Melatonin | N of 1 crossover RCT | X | | | |
| Cortesi et al[17] | 2012 Italy | ASD | Melatonin | Parallel RCT | X | X | | X |
| Van der Heijden et al[18] | 2007 Netherlands | ADHD | Melatonin | Parallel RCT | X | X | | X |
| Dodge and Wilson[23] | 2001 USA | Mixed | Melatonin | Crossover RCT | X | | | X |
| Garstang and Wallis[21] | 2006 UK | ASD with or without LD | Melatonin | Crossover RCT | X | | | |
| Hancock et al[25] | 2005 UK | Tuberous sclerosis | Melatonin | Crossover RCT | X | | | X |
| Jain et al[16] | 2015 USA | Epilepsy | Melatonin | Crossover RCT | X | X | | X |
| Jan et al[24] | 2000 Canada | Mixed | Melatonin | Crossover RCT | X | | | X |
| Wasdell et al[14] | 2008 Canada | Mixed | Melatonin | Crossover RCT | X | | | X |
| Weiss et al[15] | 2006 Canada | ADHD | Melatonin | Crossover RCT | x | X | | X |
| Wirojanan et al[19] | 2009 USA | Mixed | Melatonin | Crossover RCT | X | | | X |
| Wright et al[22] | 2011 UK | Mixed | Melatonin | Crossover RCT | X | | | X |
| **Non-pharmacological** | | | | | | | | |
| Adkins et al[28] | 2012 USA | Mixed | Parent-directed, non-tailored | Parallel RCT | X | | | |
| Austin et al[36] | 2013 Australia | Mixed | Parent-directed, tailored | Before-after | X | X | | |
| Beresford[3] | 2012 UK | Mixed | Parent-directed, tailored | Parallel RCT | X | | X | |
| Beresford[3] Intervention 2 | 2012 UK | Mixed | Parent-directed, tailored | Before-after | X | | X | |
| Beresford[3] Intervention 3 | 2012 UK | Mixed | Parent-directed non-tailored | Before-after | X | | X | |
| Beresford[3] Intervention 4 | 2012 UK | Mixed | Parent-directed non-tailored | Before-after | X | | X | |
| Bramble[60] | 1997 UK | Mixed | Parent-directed non-tailored | Before-after | X | X | X | |
| Francis and Dempster[31] | 2002 Australia | Mixed | Valerian | Crossover RCT | X | X | | |
| Gringras et al[34] | 2014 UK | Mixed | Weighted blanket | Crossover RCT | X | X | | X |
| Guilleminault et al[32] | 1993 USA | LD | Light therapy+behavioural programme | Before-after | X | | | |
| Hiscock et al[27] | 2015 Australia | ADHD plus LD or ASD | Parent-directed, tailored | Parallel RCT | X | X | X | |
| Johnson et al[26] | 2013 USA | Autism and ASD | Parent-directed, tailored | Parallel RCT | X | | | |
| Malow et al[39] | 2014 USA | Mixed | Parent-directed non-tailored | Parallel RCT | X | X | | |
| Montgomery et al[61] | 2004 UK | Mixed | Parent-directed non-tailored | Parallel RCT | X | | | |
| Moss et al[62] | 2014 Australia | Mixed | Parent-directed, tailored | Parallel RCT | X | X | X | |

**Table 1** Continued

| Study | Year Country | Type of neurodisability | Intervention | Study design | Child outcomes Sleep | Child outcomes Other | Parent outcomes | Adverse events |
|---|---|---|---|---|---|---|---|---|
| Oriel et al[35] | 2016 USA | ASD | Aquatic exercise programme | A-B-A | X | | | |
| Peppers et al[63] | 2016 USA | NR | Non-comprehensive, parent-directed | Before-after | X | X | X | |
| Piazza et al[64] | 1997 USA | Mixed | Behavioural | Parallel RCT | X | | | |
| Quine and Wade[65] | 1991 UK | LD | Parent-directed, tailored | Before-after | X | X | X | |
| Reed et al[29] | 2009 Canada | ASD | Parent-directed non-tailored | Before-after | X | X | X | |
| Sciberras et al[41] | 2011 Australia | ADHD | Parent-directed, tailored | Parallel RCT | X | X | X | |
| Weiskop et al[33] | 2005 Australia | Mixed | Parent-directed, tailored | Before-after | X | | | |
| Wiggs and Stores[30] | 1998 UK | Mixed | Non-comprehensive, parent-directed | RCT | X | X | X | |
| Yehuda et al[66] | 2011 Israel | ADHD | Essential fatty acids | Controlled before-after | X | X | | X |
| Yu et al[42] | 2015 Hong Kong | ASD and Asperger syndrome | Parent-directed non-tailored | Before-after | X | X | X | |
| Yu and Hong[67] | 2012 China | LD | Acupuncture and earpoint taping | Before-after | X | | | |

[10]Definitions for how studies were classified as parent-directed and tailored/non-tailored are available in the effectiveness review report (www.journalslibrary.nihr.ac.uk/programmes/hta/1421202/#/).

ADHD, attention deficit hyperactivity disorder; ASD, autism spectrum disorder; DD, developmental delay; LD, learning disability; NR, not reported; RCT, randomised controlled trial.

some variation. Definitions included time from lights out to sleep onset, the time from start of bedtime routine to sleep, the amount of time between when the child was put to bed and they fell asleep, and the time between taking melatonin and falling asleep.

### Other child outcomes
Nineteen studies reported child outcomes other than sleep (table 3): 15 non-pharmacological studies (58%) and four pharmacological studies (31%). The domains assessed were child behaviour, quality of life, ADHD symptoms, cognition, school-related and other. Twenty-nine different measures were used across the 18 studies and no single measure was used by more than three of the studies, though most by only one or two studies (table 3). The measures used ranged from validated tools to study-specific questionnaires.

It was not always clear whether studies had used the same version of a measure. For example, there are two versions of the Daily Parent Rating of Evening and Morning Behaviour (DREMB) scale[40]; the original version (DREMB) and a revised version (DREMB-R) with an item on irritability removed. Both studies using this outcome measure referenced the same source for the scale suggesting they used the same version. However, this could not be confirmed as only one gave details of the score range of the version used.[27 41]

Finally, we encountered a lack of clarity in reporting with respect to whether sub-scale and/or total scores were used. Thus, for the two studies which used the Child Behaviour Checklist, one reported the eight subscales scores plus a total score and daytime internalising behaviour problem score (calculated from scores on three of the subscales),[42] the other study presented insufficient information about the scores used.[39]

### Adverse outcomes
Eleven melatonin studies (85%) recorded adverse events, with variation across studies in how data were collected and reported. Different approaches were described to measuring adverse events, with varying levels of standardisation, including: a standardised assessment tool which classified events into seven domains[13]; elicitation of adverse events during in-person/phone-call visits by the study team[16 17]; using study specific questionnaires completed by parents[22 23]; one reported using a standardised form[15]; open-ended interviews[14 18]; one reported by parents using an unspecified method[25]; and in one study it was unclear how the data were collected.[19] One non-pharmacological study reported a specific method of collecting on adverse events: making a 24-hour telephone line available to parents for reporting adverse events and gathering the information at weekly parents reviews (face-to-face or telephone).[38] While other non-pharmacological studies reported some adverse events it was unclear whether there was a systematic method of gathering this information.

### Parent outcomes
Fourteen non-pharmacological studies (54%) reported parent outcomes (table 4). A range of outcome domains

**Table 2** Child sleep-related outcomes

| | Sleep outcome | Melatonin(n=13) | | Non-pharmacological (n=26) | | All studies (n=39) |
|---|---|---|---|---|---|---|
| | | Number of studies (%) | Studies measuring outcome | Number of studies (%) | Studies measuring outcome | Number of studies (%) |
| Global measures | Total sleep time | 12 (92) | 13–23 25 | 11 (42) | 26–33 35 38 39 | 23 (59) |
| | Sleep efficiency | 5 (38) | 13 14 16–18 | 5 (19) | 26–28 38 39 | 10 (26) |
| | Longest sleep episode | 1 (8) | 14 | 0 (0) | | 1 (3) |
| | Changes in sleep pattern | 1 (8) | 24 | 0 (0) | | 1 (3) |
| | CSHQ | 1 (8) | 17 | 15 (58) | 3 36 27 29 35 38 3941 42 62 63 67 | 16 (41) |
| | SBQ | 1 (8) | 16 | 0 (0) | | 1 (3) |
| | Parent-set child sleep goals | 0 (0) | | 4 (15) | 3 a, b, c, d | 4 (10) |
| | Composite Sleep Index (from modified version of Simonds and Parraga Sleep Questionnaire) | 0 (0) | | 3 (12) | 26 30 38 | 3 (8) |
| | Goal Attainment Scale | 0 (0) | | 2 (8) | 33 62 | 2 (5) |
| | Change in goal attainment rating | 0 (0) | | 1 (4) | 3 | 1 (3) |
| | Sleep problems over past 4 weeks | 0 (0) | | 1 (4) | 27 | 1 (3) |
| | Family Inventory of Sleep Habits | 0 (0) | | 3 (12) | 29 39 42 | 3 (8) |
| | Composite Sleep Disturbance Score | 0 (0) | | 1 (4) | 61 | 1 (3) |
| | Sleep quality (visual analogue scale) | 0 (0) | | 1 (4) | 31 | 1 (3) |
| | Sleep improvement – child and parent rating scale | 0 (0) | | 1 (4) | 38 | 1 (3) |
| | Quality of sleep—child smiley face rating | 0 (0) | | 1 (4) | 38 | 1 (3) |
| | Quality of sleep—Likert rating scale | 0 (0) | | 1 (4) | 66 | 1 (3) |
| Sleep initiation | Sleep onset latency | 11 (85) | 13–19 21–23 25 | 10 (39) | 26 28 29 31 33 35 36 38 39 60 | 21 (54) |
| | Bedtime | 2 (15) | 16 17 | 1 (4) | 29 | 3 (8) |
| | Sleep onset | 2 (15) | 18 19 | 1 (4) | 37 | 3 (8) |
| | Bedtime resistance | 0 (0) | | 1 (4) | 36 | 1 (3) |
| | Bedtime routine | 0 (0) | | 1 (4) | 36 | 1 (3) |
| | Number of pre-sleep disturbances | 0 (0) | | 1 (4) | 33 | 1 (3) |
| | Falling asleep in own bed | 0 (0) | | 1 (4) | 33 | 1 (3) |
| | Difficulty falling asleep (single item Likert scale) | 1 (8) | 18 | 0 (0) | | 1 (3) |
| | Severity of bedtime settling problems (Behaviour Screening Questionnaire) | | | 1 (4) | 65 | 1 (3) |
| | Time to settle at night | 0 (0) | | 2 (8) | 33 65 | 2 (5) |
| Sleep maintenance | Night awakenings | 6 (46) | 14 19 20 22 23 25 | 5 (19) | 33 35 36 38 64 | 11 (28) |
| | Number of nights without waking | 1 (8) | 20 | 0 (0) | | 1 (3) |
| | Wake time | 2 (15) | 16 18 | 2 (8) | 36 37 | 4 (10) |
| | Time spent moving during sleep period | 1 (15) | 18 | 0 (0) | | 1 (3) |
| | Co-sleeping | 0 (0) | | 2 (8) | 33 36 | 2 (5) |
| | Number of nights not sleeping in own bed | | | 1 (4) | 65 | 1 (3) |
| | Wake After Sleep Onset | 2 (15) | 16 17 | 4 (15) | 27–29 39 | 6 (15) |
| | Severity of sleep problem (VAS) (incorporating disturbed sleep) | 0 (0) | | 1 (4) | 60 | 1 (3) |
| | Movement during sleep | 0 (0) | | 1 (4) | 30 | 1 (3) |
| | Movement index | 0 (0) | | 1 (4) | 30 | 1 (3) |
| | Fragmentation index | 0 (0) | | 1 (4) | 30 | 1 (3) |
| | Time spent awake during night | 0 (0) | | 3 (12) | 31 38 65 | 3 (8) |
| | Proportion of nights with >1 wakening | 0 (0) | | 1 (4) | 38 | 1 (3) |
| | Parent/caregiver reported 'no' or 'mild' sleep problems | 0 (0) | | 1 (4) | 41 | 1 (2.6) |
| | Longest wake and sleep periods during 24 hours cycle | 0 (0) | | 1 (4) | 32 | 1 (3) |
| | Time of night wakes and return to sleep | 0 (0) | | 1 (4) | 37 | 1 (3) |
| | Hours of disturbed sleep | 0 (0) | | 1 (4) | 37 | 1 (3) |
| | Sleep Disturbance Index | 0 (0) | | 1 (4) | 36 | 1 (3) |

Continued

**Table 2** Continued

| | Sleep outcome | Melatonin(n=13) | | Non-pharmacological (n=26) | | All studies (n=39) |
|---|---|---|---|---|---|---|
| | | Number of studies (%) | Studies measuring outcome | Number of studies (%) | Studies measuring outcome | Number of studies (%) |
| Sleep scheduling | Naptime | 1 (8) | [17] | 0 (0) | | 1 (3) |
| | Napping | 0 (0) | | 1 (4) | [36] | 1 (3) |
| | Distribution of sleep bouts during 24 hours cycle | 0 (0) | | 1 (4) | [32] | 1 (3) |
| | Degree of fatigue during the day—Likert rating scale | 0 (0) | | 1 (4) | [66] | 1 (3) |
| | Interdaily stability | 1 (8) | [18] | 0 (0) | | 1 (3) |
| Other | Interdaily variability | 1 (8) | [18] | 0 (0) | | 1 (3) |
| | L5 | 1 (8) | [18] | 0 (0) | | 1 (3) |
| | Arousal Index (AASM) | 1 (8) | [16] | 0 (0) | | 1 (3) |
| | Percentage of time in each sleep stage | 1 (8) | [16] | 0 (0) | | 1 (3) |

a, intervention 1; AASM Arousal Index; b, intervention 2; c, intervention 3; CSHQ Child Sleep Habits Questionnaire; d, intervention 4; SBQ Sleep Behaviour Questionnaire; VAS Visual Analogue Scale.

were assessed, and measures used were not consistent. Domains included: parental mental health (three outcome measures across five studies), parenting (for example, perceived confidence, efficacy or knowledge in relation to managing their child's sleep problem) (five outcome measures across nine studies), quality of parental sleep (six outcome measures across three studies) and a range of other outcomes including attendance at work and locus of control (three measures across four studies). One melatonin study assessed parent outcomes; specifically, it used the Epworth Sleepiness Scale to assess parental daytime sleepiness.[13]

### Process outcomes

Several other outcomes and process measures were reported; related to adherence, feasibility of delivery, acceptability and experiences of receiving the intervention. These included: adherence to treatment (based on medication count (pharmacological studies only), intervention sessions attended, adherence to specific components of the intervention; withdrawal from study); various study-specific individual items or questionnaires to ascertain parents views about specific aspects of the intervention or overall views, for example, the proportion of participants rating treatment on a five point rating scale from 'too tough' to 'too soft'; rating of overall treatment helpfulness (visual analogue scale) and specific components of the intervention; study specific Caregivers Acceptance of Treatment Survey (Likert Scale response format); a modified version of the Program Evaluation Questionnaires and qualitative interviews. There was variation in the extent to which this information was reported in a standardised way, for example, limited detail in the methods used.

### DISCUSSION
### Principal findings

The findings suggest a strong imperative to develop a core outcome set to be used in trials in this field. Over 70 different outcome measures were reported across 39 studies, the vast majority of measures reported by three or fewer studies. This greatly impedes analysis of data in systematic reviews due to lack of comparability across studies. It also makes it difficult to assess the likelihood of selective outcome reporting, with the associated risks of overestimating intervention effectiveness. It is unlikely that all these measures have equal importance for children and parents, leading to uncertainty when making judgements about the relevance of specific study results.

There were five core areas assessed in the included studies: child sleep (six domains); other child outcomes (five domains); parent outcomes (four domains) adverse events and process measures. This suggests some consensus at the broadest level about what is relevant to measure; however, as the classification became more granular the diversity increased.

There was overlap between pharmacological and non-pharmacological studies in terms of the child sleep domains assessed and, to some extent, the measures used, reflecting the fact that, regardless of mechanism, a key target of both types of intervention was some aspect of child sleep. However, compared with the melatonin studies, there was more focus in the non-pharmacological studies on the core areas of child non-sleep outcomes, parent outcomes and family experience. This reflects the complex nature of these interventions, with parent understanding and perceived competency in managing their child's sleep being implicit in the mechanism of action. The parent outcome domains were mental health, parenting (perceived confidence, efficacy or knowledge) and quality of sleep. Arguably, it is surprising that more studies of parent-directed interventions did not assess impact on parental perceived competence/confidence and that melatonin studies have not taken as family-centred approach to assessing the effects of the intervention in this population.

The most widely used outcome measure in the melatonin studies was TST (measured using actigraphy and/or sleep diary) and in the non-pharmacological studies was the CSHQ. Although they are both within the core

**Table 3**  Other child outcomes

| | | Melatonin (n=13) | | Non-pharmacological (n=26) | | All (n=39) |
|---|---|---|---|---|---|---|
| | Outcome | Number of studies (%) | Studies measuring outcome | Number of studies (%) | Studies measuring outcome | Number of studies (%) |
| Behaviour | Daily Parent Rating of Evening and Morning Behaviour Scale | 0 (0) | | 2 (8) | 17 41 | 2 (5) |
| | Aberrant Behaviour Checklist | 1 (8) | 13 | 2 (8) | 30 38 | 3 (8) |
| | Behaviour Problem Index | 0 (0) | | 2 (8) | 60 65 | 2 (5) |
| | Child Behaviour Checklist | 1 (8) | 18 | 2 (8) | 39 42 | 3 (8) |
| | Repetitive Behaviour Scale–Revised | 0 (0)0 | | 2 (8) | 29 39 | 2 (5) |
| | Child daytime behaviour (based on diary) | 0 (0) | | 1 (4) | 31 | 1 (3) |
| | Developmental Behaviour Checklist-Parent Version | 0 (0) | | 2 (8) | 36 62 | 2 (5) |
| | Sensory Behaviour Questionnaire (unpublished) | 0 (0) | | 1 (4) | 38 | 1 (3) |
| | Behaviour Assessment System for Children-Parent Rating Scale | 1 (8) | 16 | 0 (0) | | 1 (3) |
| Quality of life | Paediatric Quality of Life Inventory 4.0 | 1 (8) | 13 | 3 (12) | 27 39 41 | 4 (10) |
| | Quality of Life in Childhood Epilepsy | 1 (8) | 16 | 0 (0) | | 1 (3) |
| | Level of good mood in general (Likert scale) | 1 (8) | 15 | 1 (4) | 66 | 2 (5) |
| | TNO-AZL Quality of Life Questionnaire | 1 (8) | 18 | 0 (0) | | 1 (3) |
| ADHD symptoms | Vanderblit ADHD Symptom Checklist | 0 (0) | | 1 (4) | 63 | 1 (3) |
| | ADHD Rating Scale IV | 0 (0) | | 2 (8) | 27 41 | 2 (5) |
| | Conner's Attention Deficit Scale–Parent Version | 1 (8) | 15 | 0 (0) | | 1 (3) |
| Cognitive | Ability to concentrate during the day, mainly at school (Likert scale) | 0 (0) | | 1 (4) | 66 | 1 (3) |
| | Working memory test battery* | 0 (0) | | 1 (4) | 27 | 1 (3) |
| | Sustained attention dots task-completion time | 1 (8) | 18 | 0 (0) | | 1 (3) |
| | Sustained attention dots task-inaccuracy† | 1 (8) | 18 | 0 (0) | | 1 (3) |
| | Erikson Task–reaction time and error incidence‡ | 1 (8) | 18 | 0 (0) | | 1 (3) |
| School-related | Homework completion (Likert scale) | 0 (0) | | 1 (4) | 66 | 1 (3) |
| | School attendance | 0 (0) | | 2 (8) | 27 41 | 2 (5) |
| | Teacher's Report Form | 1 (8) | 18 | 0 (0) | | 1 (3) |
| Other | Meal times | 0 (0) | | 1 (4) | 36 | 1 (3) |
| | Parent report of other professional help sought | 0 (0) | | 1 (4) | 27 | 1 (3) |
| | Parental Concerns Questionnaire | 0 (0) | | 1 (4) | 29 42 | 1 (3) |
| | Strength and Difficulties Questionnaire | 0 (0) | | 1 (4) | 27 | 1 (3) |
| | Parent and teacher reported core problems§ | 1 (8) | 18 | 0 (0) | | 1 (3) |

*Backwards digit recall, counting recall, listening recall.
†Percentage of misses plus false alarms relative to the total number of trials.
‡Reaction time and error incidence between congruent and incongruent tasks.
§Core problems were spontaneously and individual defined by parents.
ADHD, attention deficit hyperactivity disorder.

domain of child sleep and provide a global assessment, they differ considerably in level of objectivity, aspects of sleep assessed, feasibility and cost.[43] They could be viewed as complementary measures to be used together, though only six studies used both. Even among these more common outcome measures there was variation between studies in how they were used. Lack of standard scoring rules and variation in definitions used for actigraphy-derived outcomes has been identified as an area of concern, which is supported by our research.[44]

Adverse events were reported in the majority of melatonin trials but rarely considered in the non-pharmacological studies. Standardisation of the reporting and data collection methods for adverse events in future trials is important to understand the safety of pharmacological interventions. This also has relevance for non-pharmacological studies as well, where interventions may have unintended consequences.[45–47]

The diversity of outcome measures used significantly limited both the narrative and quantitative syntheses

**Table 4**  Parent/carer outcomes (non-pharmacological studies n=26)

| Domain | Outcome measure | Number of studies (%) | Studies measuring outcome |
|---|---|---|---|
| Parental mental health and well-being | Depression Anxiety Stress Scales | 2 (8) | 27 41 |
| | The Parenting Stress Index-Short Form | 3 (12) | 29 42 62 |
| | The Malaise Inventory | 3 (12) | 30 60 65 |
| Parenting (eg, Perceived confidence efficacy or knowledge) | Perceived ability to control own and partner's sleep difficulties (visual analogue scale) | 1 (4) | 30 |
| | Parental satisfaction with their ability to cope with their child's sleep (Likert scale) | 1 (4) | 30 |
| | Parent satisfaction with child's sleep | 1 (4) | 30 |
| | Parenting Sense of Competence scale (plus satisfaction and efficacy subscales) | 5 (19) | 3a,b,c,d 39 |
| | Knowledge of Behavioural Principles as Applied to Children | 1 (4) | 65 |
| Parental sleep | Maternal Sleep Scale | 1 (4) | 60 |
| | Pittsburgh Sleep Quality Index | 1 (4) | 42 |
| | Maternal total sleep time (actigraphy) | 1 (4) | 30 |
| | Other actigraphy measures of sleep (sleep period, activity score, movement index, fragmentation index) | 1 (4) | 30 |
| | Epworth Sleepiness Scale | 1 (4) | 30 |
| | Parental satisfaction with own sleep (Likert scale) | 1 (4) | 30 |
| Other | Work attendance (number of days missed or late for work) | 2 (8) | 27 41 |
| | Parent Satisfaction Likert Survey | 1 (4) | 63 |
| | Parental locus of Control | 1 (4) | 30 |

a, intervention 1; b, intervention 2; c, intervention 3; d, intervention 4.

conducted for the review of the effectiveness of these interventions.[9] Our findings are consistent with the results of studies in other fields exploring the extent of consistency of outcome measures used and in other evidence syntheses.[48 49]

### Unanswered questions and future research

A minimum core outcome set for use in effectiveness studies would greatly assist the usefulness of research in this field. This need not restrict researchers using outcome measures outside the core outcome set, where relevant to the specific study.[50] An agreed minimum core outcome set would improve the ability to make comparisons between studies and ensure that the outcomes being used in studies are of relevance to children, their parents and also the healthcare professionals and others involved in providing support. Although sleep has been identified as an important outcome by young people with neurodisabilities and their parents it remains unclear what aspects of sleep are most important to them, for example whether it is the child's total amount of sleep time, when they sleep, how many times they wake during the night etc. The importance to them of improving other outcomes such as child daytime behaviour or parental outcomes as part of sleep management interventions and which should be prioritised is also unclear.[51]

Effectiveness research that does not assess relevant and important outcomes is a source of avoidable waste.[52] The core outcome set should be developed in consultation with parents and carers, children themselves where possible, and healthcare professionals and others involved in supporting parents and children, using a structured process such as that developed by the COMET Group.[8] It is currently unclear whether the outcomes being assessed in studies evaluating sleep disturbance interventions are those that are important to families, both in terms of children's sleep outcomes and parents' sleep outcomes. Research in other fields has shown a disparity between the outcomes important to patients and those that are assessed in intervention studies.[49 53 54] Development of a core outcome set should be international in scope to ensure future widespread adoption. In addition to identifying core outcomes, consensus will be required regarding appropriate measurement instruments. This should draw on appropriate methods to ensure selection of reliable and valid measures such as those used by the COSMIN initiative.[55] Factors such as feasibility and respondent burden will also be important to consider. It will also be necessary to draw on existing comparative evidence on how objective and subjective sleep outcome measures perform, and the extent to which they are capturing unique dimensions of sleep.[56 57] While child-report measures exist, work on their psychometric properties is limited and none appear to have been developed for children with any significant degree of learning difficulty.[58] The heterogeneity in outcome measures used by studies in our review was compounded by the variation across studies in the clarity of information provided on how some outcomes

were measured. Future research should follow appropriate reporting guidelines.[59]

A key decision point at the outset would be whether a core outcome set should be developed separately for pharmacological and non-pharmacological interventions. We suggest that an appropriate approach would be to have a minimum core outcome set common across interventions with additional outcomes agreed where relevant for the two types of interventions. For parent-directed interventions, children's sleep outcomes are mediated by parent outcomes achieved by the intervention (for example, parental acquisition of new knowledge and understanding of sleep, training in managing sleep disturbance). Formalising and refining a theory of change for such interventions would support and inform the development of a core outcome set. This is relevant for both child and parent outcomes. Additionally, consideration is required of the variation across different types of neurodisability, where there may be desirable outcomes for one group not shared by other groups. The alternatives are to develop separate core outcome sets for children with sleep disturbance for each neurodisability or to have an overall consensus building exercise that prioritises a minimum core outcome set across conditions, with additional condition-specific outcomes identified. The latter is likely to be more feasible from a resource perspective.

Consistency in follow-up time points across studies of similar interventions, at least at the primary follow-up point, is required in order to support comparisons between studies in future systematic reviews and meta-analyses. This will require reaching consensus with regard to the most clinically meaningful follow-up time points, and whether evaluations should also seek to investigate maintenance of outcomes. Again, for non-pharmacological studies in particular, this would benefit from being based on an evidence-informed theory of change as the implementation of newly acquired knowledge and skills in managing a child's sleep may take time to have an effect. The final follow-up time-point for most studies in our effectiveness review only allowed consideration of short-term outcomes. Future studies need to consider what longer-term follow-ups should be incorporated into study designs.

### Strengths and weaknesses of the study

The systematic review included a wide range of neurodisabilities and interventions in children aged 18 years or under. It therefore provides a comprehensive overview of the outcome domains assessed in this field of research and the outcome measures being used. However, we may have under-estimated the range of outcomes being assessed as this work was undertaken as part of a systematic review of effectiveness. Thus, although the outcomes of interest were very broad we did not exhaustively include all outcomes; therefore, we may have failed to identify some outcomes, for example some physiological aspects of sleep. In addition, we assigned the measures to one of a number of outcome domains, others doing the same task

may allocate differently or attribute to different domains. However, these factors are unlikely to change the conclusions. The description of outcome domains and measures used provides important evidence for the field in moving towards developing a core outcome set. While these will contribute to the development of a list of outcomes for a Delphi survey, as a first stage of developing a core outcome set, it is important that this is supplemented by the views of parents and children, healthcare professionals involved in designing and delivering interventions and researchers working in the field.

### CONCLUSION

There is a lack of consistency between studies in the outcome measures used to assess the effectiveness of interventions for non-respiratory sleep disturbances in children with neurodisabilities. This hampers evidence synthesis and creates uncertainty about the relevance of study findings to parents and children. A minimum core outcome set, with international consensus, should be developed in consultation with parents, children and young people, and those involved in supporting families.

**Author affiliations**
[1]York Trials Unit, Department of Health Sciences, University of York, York, UK
[2]The Children's Sleep Charity, Doncaster, UK
[3]Department of Respiratory Medicine, Sheffield Children's NHS Foundation Trust, Sheffield, UK
[4]Newcastle University, Newcastle upon Tyne, UK
[5]Institute of Health and Society, Blackpool Teaching Hospitals NHS Foundation Trust, Blackpool, UK
[6]Children's and Adolescent Services, Social Policy Research Unit, University of York, York, UK

**Acknowledgements** We would like to thank our parent advisors for their interest and enthusiasm for the project and their contributions at project meetings.

**Contributors** CMD: jointly responsible for writing the protocol and had shared responsibility for co-ordinating and leading the project; provided advice and input to all elements of the project. Wrote sections of main report and wrote and revised drafts of this paper. AP and AS: equally contributed to study selection, data extraction, quality assessment, and synthesis and report writing of the main report. Commented on drafts of this paper. CF: conducted the data analysis, contributed to the report writing and commented on drafts of this paper. CH: contributed to the protocol, provided methodological advice throughout the project, commented on drafts of the main report and drafts of this paper. GS: worked on the project February–September 2016. During that time she was responsible for the day-to-day running of the project and led on screening, retrieval, data extraction and quality appraisal. Commented on drafts of this paper. VD, HE, MT: provided expert clinical advice throughout the project, contributed to screening and data extraction processes and to drafts of the report, commented on drafts of this paper. BB: jointly responsible for writing the review protocol and had shared responsibility for co-ordinating and leading the project; provided advice and input to all elements of the project. Wrote sections of main report and commented on drafts of this paper.

**Funding** This work is derived from a project funded by the National Institute for Health Research (NIHR) HTA Programme (project number 14/212/02). Further information is available at https://www.journalslibrary.nihr.ac.uk/programmes/hta/1421202/#/. This report presents independent research commissioned by the NIHR.

**Disclaimer** The views and opinions expressed by authors in this publication are those of the authors and do not necessarily reflect those of the NHS, the NIHR, MRC, CCF, NETSCC, the HTA Programme or the Department of Health.

**Competing interests** BB and MT were authors of studies included in the review, but were not involved in data extraction or quality assessment of these studies.

CM is a member of the NIHR HTA & EME Editorial Board. CH is a member of HTA Commissioning Board

**Patient consent for publication** Not required.

**Provenance and peer review** Not commissioned; externally peer reviewed.

**Data sharing statement** The data are derived from published studies and are publicly available in the main body and appendices of the HTA Monograph.

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
