## [Reviewer comments · BMJ Open]

ARTICLE DETAILS

TITLE (PROVISIONAL)	Outcome domains and outcome measures used in studies assessing the effectiveness of interventions to manage non-respiratory sleep disturbances in children with neurodisabilities: a systematic review
AUTHORS	McDaid, Catriona; Parker, Adwoa; Scantlebury, Arabella; Fairhurst, Caroline; Dawson, Vicky; Elphick, Heather; Hewitt, Catherine; Spiers, Gemma; Thomas, Megan; Beresford, Bryony

VERSION 1 - REVIEW

REVIEWER	Helen S Heussler Children's Health Queensland, Australia
REVIEW RETURNED	11-Nov-2018

GENERAL COMMENTS	This is a useful paper in terms of the tools currently being utilised in non respiratory sleep management however there are some points that should be addressed particularly in the area of the limitations of this study in its application for wider trials. This might be a personal view but I believe that this is very helpful in terms of domains but urge some caution in promoting a defined list of "tests" as it stifles the researcher to address novel aspects This is a challenging area - particularly in the area of disability with the significant heterogeneity of the populations and one size might not fit all It would be useful to include the measures and what they were measured by in the table- ie Actigraphy or parental or child report- we know that these diverge significantly after the age of 9 years and that the reliability of actigraphy in the population with disabilities is very variable. In the limitations you should discuss the challenges in assessing the sleep of a child who is unable to communicate the subjective nature of sleep vs the objective nature I like the specific domains(which I think you have done nicely (I suspect the different tools are a bit country specific) - rather than specific questionnaires from this study but it is helpful to see the variety. Nominating specific Questionnaires usage and monopoly can result in publication biases and it may be worth mentioning cautions around this. It would also be worth commenting on the fact that for children with disabilities the same measures are not always very useful as in the typically developing population such
--

	as anxiety. It reads a little like this is the aim although I think this isn't the intent. Challenges around dictation of outcomes can be problematic and miss the true effectiveness- so please be cautious in this area. Consistency of some domain specific areas is important for some trials but will differ depending on what you are trialling (eg something that might be aimed to shift Circadian rhythm vs something that is inclined to sedate) I suspect so maybe a sentence or two about this would help. I think if these concerns are addressed this is reasonable to be published.
--	---

REVIEWER	Colin Reilly Department of Paediatrics Queen Silvias Childrens' Hospital Gothenburg Sweden
REVIEW RETURNED	17-Nov-2018

GENERAL COMMENTS	I think that this is well written. I enjoyed Reading it and feel that the authors have done a good job. - I have a few comments which may be of help to the authors. Abstract Children - i thought Children igh be defined as 17 years and younger and not 18 but this is just a thought efficacy or understanding - does this refer to parents - perhaps it could be made clearer Page 5 - Full search strategies re available from the corresponding author - could this not be part of supplementary data? I think it would be nice to define neurodisability in the paper and not just refer to Morris et al. Its not a term universally used so i think it should perhaps be defined in the paper if possible. Page 7 - perhaps write fourteen as 14 Page 11 - in the table is it possible to write what disorders were involved i.e. was it autism, epilepsy, a mixed Group? Discussion - i think it is important to at least discuss the possibility that outcome measures might differ across conditions. In my own field of neurology parents of Children with epilepsy may have seizure related needs or desired outcomes not shared by Children with other conditions.
--

REVIEWER	Dr Ibtihal Abdelgadir Sidra Medicine & Weill Cornell Medical School Doha Qatar
REVIEW RETURNED	01-Dec-2018

GENERAL COMMENTS	Dear authors, Thank you for submitting this well written systematic review for consideration of publication.
--

	To match the high standard of writing of this manuscript, please consider these points: 1. Data screening done by one researcher screened titles only to remove obviously irrelevant studies (10% checked by a second researcher) The standard method in data screening to be done by two researchers to minimize bias. 2. Data extraction and synthesis Data were extracted by one researcher and checked by a second. The standard method in data extraction and synthesis is to be done by two researchers to minimize bias. Kind regards
--	--

VERSION 1 – AUTHOR RESPONSE

Reviewer 1

This is a useful paper in terms of the tools currently being utilised in non respiratory sleep management however there are some points that should be addressed particularly in the area of the limitations of this study in its application for wider trials. This might be a personal view but I believe that this is very helpful in terms of domains but urge some caution in promoting a defined list of "tests" as it stifles the researcher to address novel aspects. This is a challenging area - particularly in the area of disability with the significant heterogeneity of the populations and one size might not fit all	Thank-you for raising this point as we realise we have not been clear about this in the discussion. We have revised the discussion to make clearer that by core outcome set we mean a standardised set of outcomes, which should be used as a minimum – this would not prevent researchers from measuring other outcomes they thought relevant to the intervention under investigation. We have also added “minimum” to the conclusion in the abstract and main body of paper. We have also added a point to the discussion about whether different types of neurodisabilities need to be considered separately.
It would be useful to include the measures and what they were measured by in the table- ie Actigraphy or parental or child report- we know that these diverge significantly after the age of 9 years and that the reliability of actigraphy in the population with disabilities is very variable.	This has been added to the text as it cannot be added to the table, which provides aggregate data and not by study.
In the limitations you should discuss the challenges in assessing the sleep of a child who is unable to communicate the subjective nature of sleep vs the objective nature	We have added further text to the discussion to address this issue as requested.
I like the specific domains(which I think you have done nicely (I suspect the different tools are a bit country specific) - rather than specific questionnaires from this study but it is helpful to see the variety.	Thank-you for the comment
Nominating specific Questionnaires usage and monopoly can result in publication biases and it may be worth mentioning cautions around this. It would also be worth commenting on the fact	Thanks for this comment which we have addressed as described above

that for children with disabilities the same measures are not always very useful as in the typically developing population such as anxiety. It reads a little like this is the aim although I think this isn't the intent. Challenges around dictation of outcomes can be problematic and miss the true effectiveness- so please be cautious in this area. Consistency of some domain specific areas is important for some trials but will differ depending on what you are trialling (eg something that might be aimed to shift Circadian rhythm vs something that is inclined to sedate) I suspect so maybe a sentence or two about this would help.	
---	--

Reviewer 2

I think that this is well written. I enjoyed Reading it and feel that the authors have done a good job. I have a few comments which may be of help to the authors.	Thank-you for the comment
I thought Children igh be defined as 17 years and younger and not 18 but this is just a thought	The boundaries of the review were set to be inclusive, as we were commissioned to provide a broad overview of the evidence. Several primary studies set their inclusion criteria as children \leq 18 years so our criteria reflect practice in the field. In reality the mean age in studies was much younger than the upper age limit.
Full search strategies re available from the corresponding author - could this not be part of supplementary data?	The ASSIA search has been provided as a supplementary file.
I think it would be nice to define neurodisability in the paper and not just refer to Morris et al. Its not a term universally used so i think it should perhaps be defined in the paper if possible.	This has been added to the paper.
Page 7 - perhaps write fourteen as 14	This has been changed.
Page 11 - in the table is it possible to write what disorders were involved i.e. was it autism, epilepsy, a mixed Group?	This information has been added to the table
Discussion - i think it is important to at least discuss the possibility that outcome measures might differ across conditions. In my own field of neurology parents of Children with epilepsy may have seizure related needs or desired outcomes not shared by Children with other conditions.	Thanks for drawing our attention to this – the heterogeneity of the group may be an important consideration in deciding how to proceed with a core outcome set - we have added this as a point to the discussion

Reviewer 3

Thank you for submitting this well written systematic review for consideration of	Thank-you for the comment
---	---------------------------

publication. To match the high standard of writing of this manuscript, please consider these points:	
Data screening done by one researcher screened titles only to remove obviously irrelevant studies (10% checked by a second researcher) The standard method in data screening to be done by two researchers to minimize bias.	Following removal of obviously irrelevant titles and abstracts, the remaining titles and abstracts were independently screened by two researchers and full papers were screened independently by two researchers.
Data extraction and synthesis Data were extracted by one researcher and checked by a second. The standard method in data extraction and synthesis is to be done by two researchers to minimize bias	While duplicate data extraction is the gold standard, extraction by one researcher and checking by a second is an accepted approach to minimising risk of error and bias, which is in accordance with the Centre for Reviews and Dissemination's Guidance for Undertaking Reviews in Healthcare

VERSION 2 – REVIEW

REVIEWER	Dr Ibtihal Abdelgadir Sidra Medicine Qatar
REVIEW RETURNED	14-Mar-2019
GENERAL COMMENTS	Thank you for your submission of the revised manuscript.